# Anti-TNF Biologicals Enhance the Anti-Inflammatory Properties of IgG N-Glycome in Crohn’s Disease

**DOI:** 10.3390/biom13060954

**Published:** 2023-06-07

**Authors:** Maja Hanić, Frano Vučković, Helena Deriš, Claire Bewshea, Simeng Lin, James R. Goodhand, Tariq Ahmad, Irena Trbojević-Akmačić, Nicholas A. Kennedy, Gordan Lauc, PANTS Consortium

**Affiliations:** 1Genos Glycoscience Research Laboratory, 10000 Zagreb, Croatia; mhanic@genos.hr (M.H.); fvuckovic@genos.hr (F.V.); hderis@genos.hr (H.D.); iakmacic@genos.hr (I.T.-A.); 2Exeter Inflammatory Bowel Disease and Pharmacogenetics Research Group, University of Exeter, Exeter EX4 4SB, UK; claire.bewshea@nhs.net (C.B.); simeng.lin1@nhs.net (S.L.); james.goodhand@nhs.net (J.R.G.); tariq.ahmad1@nhs.net (T.A.); nick.kennedy1@nhs.net (N.A.K.); 3Faculty of Pharmacy and Biochemistry, University of Zagreb, 10000 Zagreb, Croatia

**Keywords:** Crohn’s disease, IgG glycosylation, infliximab, adalimumab, PANTS study

## Abstract

Crohn’s disease (CD) is a chronic inflammation of the digestive tract that significantly impairs patients’ quality of life and well-being. Anti-TNF biologicals revolutionised the treatment of CD, yet many patients do not adequately respond to such therapy. Previous studies have demonstrated a pro-inflammatory pattern in the composition of CD patients’ immunoglobulin G (IgG) N-glycome compared to healthy individuals. Here, we utilised the high-throughput UHPLC method for N-glycan analysis to explore the longitudinal effect of the anti-TNF drugs infliximab and adalimumab on N-glycome composition of total serum IgG in 198 patients, as well as the predictive potential of IgG N-glycans at baseline to detect primary non-responders to anti-TNF therapy in 1315 patients. We discovered a significant decrease in IgG agalactosylation and an increase in monogalactosylation, digalactosylation and sialylation during the 14 weeks of anti-TNF treatment, regardless of therapy response, all of which suggested a diminished inflammatory environment in CD patients treated with anti-TNF therapy. Furthermore, we observed that IgG N-glycome might contain certain information regarding the anti-TNF therapy outcome before initiating the treatment. However, it is impossible to predict future primary non-responders to anti-TNF therapy based solely on IgG N-glycome composition at baseline.

## 1. Introduction

Inflammatory bowel disease (IBD) is a chronic immune-mediated inflammatory disease of the gastrointestinal tract that can be classified as Crohn’s disease (CD) or ulcerative colitis (UC) using available diagnostic tools. In 2017, nearly 3.9 million women and 3.0 million men worldwide were affected by IBD [1]. The prevalence of IBD continues to increase globally, and it significantly burdens healthcare systems and economies, particularly in developing countries [1,2]. Although the aetiology of IBD is still not fully understood, it is generally considered that aberrant mucosal innate and adaptive immune responses [3] and environmental risk factors, such as early exposure to antibiotics, poor diet, air pollution, psychological stress and altered composition of gut microbiota [4,5], significantly contribute to the pathogenesis of IBD in genetically susceptible individuals [6]. However, protein glycosylation is another critical component associated with the development and progression of IBD [7,8]. Glycans are a vast group of complex oligosaccharides found on proteins and lipids in our bodies. Their monosaccharide composition is tailored by genetic and environmental influences [9], making them an excellent reflection of the body’s current state with exquisite diagnostic, prognostic and biomarker potential [9]. Alterations in the glycome (set of glycans) of immunoglobulin G (IgG), the central molecule of humoral immunity, has been extensively studied in ageing, carcinomas and inflammatory diseases such as autoimmune diseases and chronic inflammatory states (e.g., low back pain) [10]. Biantennary N-glycans of IgG are responsible for fine-tuning its effector functions by modulating the interaction of the fragment crystallisable (Fc) part of the IgG molecule with various Fcγ receptors on effector cells and, consequently, orchestrating the activity of complement-dependent cytotoxicity (CDC) and antibody-dependent cell-mediated cytotoxicity (ADCC) [11,12,13]. In addition, roughly 20% of IgG molecules are glycosylated in the variable region of the fragment antigen-binding (Fab) domain, which is essential for the interaction of antibodies with specific antigens and its immunomodulatory effect through interaction with lectins [14]. Decreased galactosylation of IgG is a hallmark of many inflammatory diseases [10], and IBD is no exception. In fact, the IgG N-glycome shows increased pro-inflammatory potential in CD patients compared to healthy individuals. The IgG N-glycan repertoire in CD contains fewer galactosylated and sialylated N-glycans and more glycans with bisecting *N*-acetylglucosamine (GlcNAc) [15,16], and it can also distinguish CD patients from healthy controls and UC patients [16]. 

Inflammation in CD is promoted by increased intestinal tissue levels of tumour necrosis factor (TNF), along with interferon-gamma and interleukin (IL)-12, all of which are potent pro-inflammatory cytokines [17,18]. Early and effective control of inflammation in CD patients is of utmost importance. The treatment of CD reached a significant milestone when the first anti-TNF drug, the chimeric IgG1 monoclonal antibody infliximab (IFX), was introduced into practice in 1998 [19]. Both IFX and the fully human IgG1 monoclonal antibody adalimumab (ADA) are now widely used for CD. Their mechanism of action involves Fab-mediated neutralisation of transmembrane and soluble TNF and destruction of TNF-producing cells through CDC and ADCC by the interaction of the IgG Fc region with receptors on effector cells [20]. This interaction is strongly influenced by N-glycans bound to the IgG Fc region [21,22].

There are several reports of anti-TNF therapy changing the serum and IgG glycome composition in chronic inflammatory diseases (CID) [16,23,24]. However, knowledge about the longitudinal influence of anti-TNF treatment on IgG glycome in CD patients is relatively scarce. 

Up to one-third of patients do not respond to anti-TNF therapy induction regimen (primary non-response, PNR), and a further 30–50% of patients who initially respond to therapy lose their response during the maintenance regimen (secondary loss of response, LOR) [25,26]. Prediction of therapy response to anti-TNF is one of the ultimate goals in the personalised approach to treating IBD. However, all the predictive biomarkers reported to date lack clinical utility [25]. In patients with UC, low levels of branched glycans in intestinal T cells associate with failure of standard therapy independent of other clinical parameters. The predictive accuracy is further increased when the C-reactive protein (CRP) level is accounted [27]. 

Therefore, this study aims to investigate (A) longitudinal changes in IgG N-glycome in CD patients treated with anti-TNF monoclonal antibodies IFX and ADA, and (B) baseline IgG N-glycome patterns that predict primary non-response to IFX and ADA.

## 2. Materials and Methods

### 2.1. Clinical Samples and Ethical Considerations

Patients with CD were recruited as a part of the Personalized Anti-TNF Therapy in Crohn’s Disease (PANTS) study conducted in hospitals across the United Kingdom. The PANTS study was a prospective uncontrolled observational cohort study investigating the mechanism of PNR, LOR and adverse drug reactions (ADR) to IFX and ADA in anti–TNF-naïve patients with severe active luminal CD. To enter the study, subjects had to meet several inclusion criteria: age of 6 years and over, presence of active luminal CD involving the colon and/or small intestine (Montreal classification L1, L2 or L3) supported with raised CRP and/or faecal calprotectin levels, and no prior exposure to anti-TNFα medication. A total of 1513 CD patients’ serum samples were collected at two time points, before the first dose of anti-TNF therapy (week 0/baseline, N = 1315) and for subset of patients, immediately before the next scheduled anti-TNF injection/infusion (week 14, N = 198). PNR was determined at week 14 by the following: ongoing use of corticosteroids, cessation of anti-TNF drug for non-response, or failure of both the Harvey-Bradshaw Index (HBI) to fall by 3 or more points from week 0 baseline or to 4 or below, and the CRP to fall to within normal range (≤3 mg/L) or by 50% from week 0. Grey zone, as an intermediate response between PNR and response, was defined as a decrease in CRP level to 3 mg/L or less, or by 50% or more from baseline; or as a decrease in HBI score to 4 points or less, or by 3 points or more from baseline; but not both. Treatment response was defined as a decrease in CRP to 3 mg/L or less, or by 50% or more from baseline and a decrease in HBI score to 4 points or less, or by 3 points or more from baseline for adults; or a decrease in sPCDAI to 15 points or less, or by 12.5 points from baseline for children. Remission was defined as CRP of 3 mg/L or less and HBI score of 4 points or less (sPCDAI score ≤ 15 points), no ongoing steroid therapy, and no exit due to treatment failure. The patient’s gastroenterologist decided to continue or suspend anti-TNF therapy between weeks 12 and 14. Samples were collected with the approval of The South West Research Ethics Committee (Research Ethics Committee reference: 12/SW/0323) and informed written consent was obtained from all participants.

### 2.2. IgG N-Glycan Sample Preparation

Prior to IgG isolation, block randomisation was performed to define the position of samples across 96-well plates, and replication standard samples were included as well. Sample preparation was based on previously described protocols [28,29] for hydrophilic-interaction-based ultra-high performance hydrophilic liquid chromatography with fluorescence detection (HILIC-UHPLC-FLD) for IgG N-glycan analysis in a high-throughput manner. Briefly, 100 µL of serum was diluted with 700 µL 1 × PBS and applied to a Protein G monolithic plate (BIA Separations, Ajdovscina, Slovenia). After three 1 × PBS washes, IgG was eluted in 1 mL 0.1 M formic acid (Merck, Darmstadt, Germany) and neutralised with 170 µL 1M ammonium bicarbonate (Merck). An equal volume of each IgG eluate containing an equivalent of 100–300 µg of IgG was dried in a vacuum concentrator and denatured with 30 µL of 1.33% (*w*/*v*) SDS (Invitrogen, Carlsbad, CA, USA) by incubation at 65 °C for 10 min. Denatured IgG was incubated with 10 µL of 4% (*v*/*v*) Igepal-CA630 (Sigma-Aldrich, St. Louis, MO, USA) and deglycosylated by addition of 1.2 U of PNGase F (Promega, Madison, WI, USA) in 10 µL of 5 × PBS for 18 h at 37 °C. Released N-glycans were labelled with a labelling mixture. The labelling mixture was freshly prepared by dissolving 0.48 mg 2-aminobenzamide (2-AB, Sigma-Aldrich) and 1.12 mg 2-picoline borane (2-PB, Sigma-Aldrich) in 25 µL of dimethyl sulfoxide (DMSO, Sigma-Aldrich) and glacial acetic acid (Sigma-Aldrich) mixture (70:30, *v*/*v*) per sample. After a 10 min shake, a labelling reaction was conducted for two hours at 65 °C. The free label and reducing agent excess were removed by hydrophilic interaction liquid chromatography-solid phase extraction (HILIC-SPE). For that purpose, 2-AB labelled IgG N-glycans (total volume of 75 µL) were diluted with 700 µL of acetonitrile (ACN) and applied to an AcroPrep GHP filter plate with 0.2 µm pore diameter (Pall Corporation, Ann Arbor, MI, USA). After five washes with 96% (*v*/*v*) ACN using a vacuum manifold, 2-AB labelled IgG glycans were collected in two fractions of 90 µL ultra-pure water (total volume of 180 µL).

### 2.3. HILIC-UHPLC-FLD Analysis of 2-AB Labelled IgG N-Glycans

Fluorescently labelled N-glycans were separated by hydrophilic interaction liquid chromatography on a Waters Acquity Ultra performance liquid chromatography (UPLC) H-class instrument (Waters Corporation, Milford, MA, USA) consisting of a quaternary solvent manager, sample manager and a fluorescence detector set with excitation and emission wavelengths of 250 and 428 nm, respectively. The instrument was under the control of Empower 3 software, build 3471 (Waters Corporation). Labelled N-glycans were separated on a Waters bridged ethylene hybrid (BEH) 100 mm × 2.1 mm glycan chromatography column filled with 1.7 μm BEH particles. The mobile phase consisted of 100 mM ammonium formate (pH 4.40) as solvent A and ACN as solvent B. The separation method used a linear gradient of 75% to 62% ACN (*v*/*v*) at a flow rate of 0.4 mL/min in a 28 min analytical run. Samples were maintained at 10 °C before injection, and the column temperature was set at 60 °C. In order to calibrate UHPLC runs against day-to-day and system-to-system changes, an external standard of hydrolysed and 2-AB labelled glucose oligomers (dextran ladder) was used as a reference from which the retention times for the individual glycans were converted to glucose units (GU). Obtained chromatograms were all separated in the same manner into 24 glycan peaks (GP1–GP24) via the automatic chromatogram extraction (ACE) method [30]. The amount of glycans in each peak was expressed as a percentage (%) of the total integrated area.

### 2.4. Statistical Analysis

In order to remove experimental variation from measurements, normalisation and batch correction were performed on UHPLC IgG N-glycan data. To make measurements across samples comparable, normalisation by total area was performed, where the peak area of each of 24 glycan structures was divided by the total integrated area of the corresponding chromatogram. Before the batch correction, normalised glycan measurements were log-transformed due to right-skewness of their distributions and the multiplicative nature of batch effects. Batch correction was performed on log-transformed measurements using ComBat method (R package sva), where the technical source of variation (which sample was analysed on which plate) was modeled as a batch covariate. Estimated batch effects were subtracted from log-transformed measurements to get measurements corrected for experimental noise. An additional six derived traits were calculated from 24 directly measured and normalised glycan traits (GP1-GP24) and defined as: the percentage of agalactosylated glycans in total IgG glycans–G0 = SUM(GP1 + GP2 + GP3 +GP4 + GP6)/GP × 100; the percentage of monogalactosylated glycans in total IgG glycans–G1 = SUM(GP7 + GP8 + GP9 +GP10 + GP11)/GP × 100; the percentage of digalactosylated glycans in total IgG glycans–G2 = SUM(GP12 + GP13 + GP14 + GP15)/GP × 100; the percentage of mono- and disialylated glycans in total IgG glycans–S = SUM(GP16 + GP17 + GP18 + GP19 + GP21 + GP22 + GP23 + GP24)/GP × 100; the percentage of glycans with core fucose in total IgG glycans–F = SUM(GP1 + GP4 + GP6 + GP8 + GP9 + GP10 + GP11 + GP14 + GP15 + GP16 + GP18 + GP19 + GP23 + GP24)/GP × 100; the percentage of glycans with bisecting *N*-acetylglucosamine (GlcNAc) in total IgG glycans–B = SUM(GP3 + GP6 + GP10 + GP11 + GP13 + GP15 + GP19 + GP22 + GP24)/GP × 100. These derived traits average particular glycosylation features across different individual glycan structures and therefore are more closely related to individual enzymatic activities and underlying genetic polymorphisms. 

Longitudinal analysis of patient samples through their observation period was performed by implementing a linear mixed effects model where glycan measurement was the dependent variable, time was modeled as a fixed effect while individual ID was included in a model as a random intercept, with age and gender included as additional covariates. Treatment effect on N-glycome change through time was analysed using a linear mixed effects model where time was modelled as a fixed effect, the interaction between time and treatment was modelled as a fixed effect, while individual sample ID was modeled as a random intercept, with age and gender included as additional covariates. Association analyses between therapy response/remission status and baseline glycomic measurements were performed using a regression model with age, gender, BMI, disease duration, disease location and behaviour included as additional covariates. Prior to analyses, glycan variables were all transformed to a standard normal distribution (mean = 0, standard deviation = 1) by inverse transformation of ranks to normality (R package “GenABEL”, function rntransform). Using rank-transformed variables in analyses makes estimated effects of different glycans comparable as transformed glycan variables have the same standardised variance. False discovery rate was controlled using Benjamini-Hochberg procedure (function p.adjust(method = “BH”)). Data were analysed and visualised using R programming language (version 4.0.2).

## 3. Results

Using HILIC-UHPLC-FLD as a high-throughput method for IgG N-glycan analysis, we successfully glycoprofiled and quantified a total of 1513 IgG samples isolated from the serum of CD patients treated with IFX or ADA monoclonal antibodies from the PANTS cohort. More detailed demographic characteristics of included CD patients are given in Table 1. The IgG N-glycoprofiles obtained from each patient were separated into 24 glycan peaks (GP1-GP24), the composition of which has been reported previously (Appendix A) [31]. Relative glycan abundance in a particular glycan peak was expressed as a percentage of the total integrated area. Furthermore, glycans with shared structural features were summarised into six derived traits and included in the statistical analysis. We compared the IgG N-glycan profiles of 198 patients whose serum samples were longitudinally collected at baseline (week 0) and at week 14 to assess the effect of anti-TNF therapy on the composition of IgG N-glycome of CD patients. Next, we searched for patterns in the IgG N-glycome of 1315 CD patients whose serum samples were collected before anti-TNF induction to identify biomarkers of primary non-response. 

### 3.1. Anti-TNF Therapy Changes the IgG N-Glycome Composition of Crohn’s Disease Patients

Overall, we identified several significant changes in the levels of derived IgG N-glycan traits between baseline and week 14 of treatment with both IFX and ADA (Figure 1). These changes were observed in CD patients regardless of their response to treatment. The most pronounced change discovered was the decreased abundance of IgG N-glycans lacking galactose (agalactosylation, G0). This change was followed by an increase in the levels of monogalactosylated (G1), digalactosylated (G2) and sialylated glycans (S). However, no statistically significant change was discovered in glycans with core-fucose (F) or bisecting GlcNAc (B) (Table 2). Results for the individual directly measured glycan traits are given in Appendix A. 

### 3.2. Infliximab Causes More Pronounced Changes in IgG N-Glycome than Adalimumab in Crohn’s Disease Patients

We compared the group of patients treated with ADA (n = 98) with a group treated with IFX (n = 100). Both drugs change the composition of IgG N-glycome in the same direction. In other words, the abundance of G0 glycans decreased, and G1, G2 and S glycans increased in their level in the course of 14 weeks, while B and F derived traits remained the same in that particular period (Figure 2). However, IFX caused a significantly greater decrease in the G0 trait and a greater increase in G1, G2 and S derived traits in CD patients compared to the group treated with ADA (Table 3).

### 3.3. Composition of IgG N-Glycome Differs between Groups with Different Anti-TNF Therapy Outcomes

Baseline measurements of IgG N-glycans expressed through derived traits were compared between the four groups with different therapy outcomes established at week 14 (PNR, response; grey zone, remission) (Figure 3). A statistically significant association was observed between therapy outcomes and G0, G1, G2 and S derived IgG N-glycan traits in general (Table 4). However, it is impossible to clearly distinguish the potential predictors of therapy non-response (PNR) based only on the composition of IgG N-glycome at baseline nor observe any particular change trend between the four groups. Pairwise associations of derived IgG N-glycan traits at baseline with therapy outcome are given in Appendix A.

## 4. Discussion

Crohn’s disease is a type of IBD that can affect individuals of any age and may result in severe disability and morbidity since most individuals will develop complications or require surgery within ten years of diagnosis [32]. Therefore, early diagnosis, proper disease control, and, most importantly, induction and remission maintenance are crucial in managing CD. Anti-TNF drugs revolutionised the treatment of CD. However, up to 40% of patients do not respond to therapy (PNRs), almost half of the patients may have a secondary LOR, and roughly 10% may develop adverse drug reactions [33]. Due to its integral role in humoral immune processes, IgG is one of the most studied glycoproteins in health and disease, and its glycan part plays a vital role in the modulation of IgG effector functions [10]. Several studies reported lower overall galactosylation [15,16,23,34] and a decrease in sialylation [15,16,23] of IgG glycome in CD patients compared to the healthy controls, indicating an enhanced pro-inflammatory potential of serum IgG in CD. Here, we have successfully glycoprofiled the total serum IgG of 198 CD patients to examine the effect of anti-TNF monoclonal antibodies IFX and ADA on the IgG N-glycome during the first 14 weeks of the treatment. We discovered a significant shift in IgG N-glycome composition towards less inflammatory glycosylation patterns regardless of therapy response to IFX and ADA. To elaborate, the relative abundance of agalactosylated glycans was significantly lower 14 weeks after the initiation of anti-TNF therapy. The observed decrease in agalactosylated glycans was accompanied by an increase in monogalactosylated and digalactosylated glycans and an increase in sialylated glycans. At the same time, no significant changes were observed for fucosylated glycans nor glycans with bisecting GlcNAc. These results are accordant to the recently reported longitudinal changes in patients with various chronic inflammatory diseases such as CD, UC, systemic lupus erythematosus (SLE) and arthritic patients treated with anti-TNF therapy as well [23]. 

Our subsequent finding demonstrated an increased capability of IFX to change the IgG N-glycome towards a less inflammatory pattern compared to ADA. Several previous findings suggested that IFX is more effective in the induction of remission in CD patients than ADA [35,36] and in providing a clinical response, mucosal healing and clinical remission in UC patients [37]. It has also been reported that IFX effectively reduced the concentration of agalactosylated IgG in RA [38] and inflammatory arthritis [24], and which was also associated with clinical improvement for those patients. 

It is still not fully understood how anti-TNF therapy affects the concentration of primarily agalactosylated IgG glycoforms and dampens systemic inflammation. However, it is evident that both IFX and ADA have an opposite effect on the IgG N-glycome composition compared to the effect of CD, and it should be further investigated in more detail. 

Many studies report that galactosylated N-glycans enhance the anti-inflammatory properties of total serum IgG, as seen in RA patients and pregnancy-induced improvement, where enhanced galactosylation accompanied by lower agalactosylation of total serum IgG was associated with clinical improvement [39], independent of sialylation [40]. Even though the role of sialylation as a switch between pro- and anti-inflammatory properties of IgG is still a matter of debate, it is considered that the presence of sialylated N-glycans contributes to the anti-inflammatory properties of IgG by diminishing IgG affinity for activating receptor FcyRIIIA, reducing ADCC [10] and impairing CDC by decreasing C1q binding [41]. 

Since therapy response prediction is one of the ultimate requirements in the era of personalised medicine, we searched for differences in the IgG N-glycome composition of 1315 CD patients at baseline that might identify PNR. Patients were divided into four groups (PNR, response, grey zone and remission) based on the therapy outcome established at week 14. We discovered that information about therapy response/outcome at week 14 might be contained in the composition of IgG glycome at baseline. However, even though a statistically significant association between the derived IgG N-glycan traits and different therapy outcomes was detected, we could not specify which group significantly differed from another, nor could we observe any trend of change when focusing on transitioning from one therapy outcome to another. The obtained results suggest that IgG N-glycan composition at baseline does not have a predictive potential to detect future PNRs or other anti-TNF therapy outcomes. Similarly, a recently published paper from our group discovered that it is not possible to provide information about future disease activity based solely on IgG Fc glycan composition at baseline. However, the distinction between the patients in remission and those with active CD was instead contained in the rate of change of IgG Fc N-glycome [23].

In conclusion, we replicated and expanded the knowledge about the effect of longitudinal use of anti-TNF therapy IFX and ADA on the composition of total serum IgG N-glycome in patients with active luminal CD in a large, well-described, prospective, uncontrolled observational PANTS cohort. Even though we detected significant differences in the relative abundance of derived IgG N-glycan traits between the groups with different therapy outcomes, the composition of IgG N-glycome at baseline does not hold a significant predictive power to detect PNRs among CD patients. Therefore, further studies are warranted in this direction.

## Figures and Tables

**Figure 1 biomolecules-13-00954-f001:**
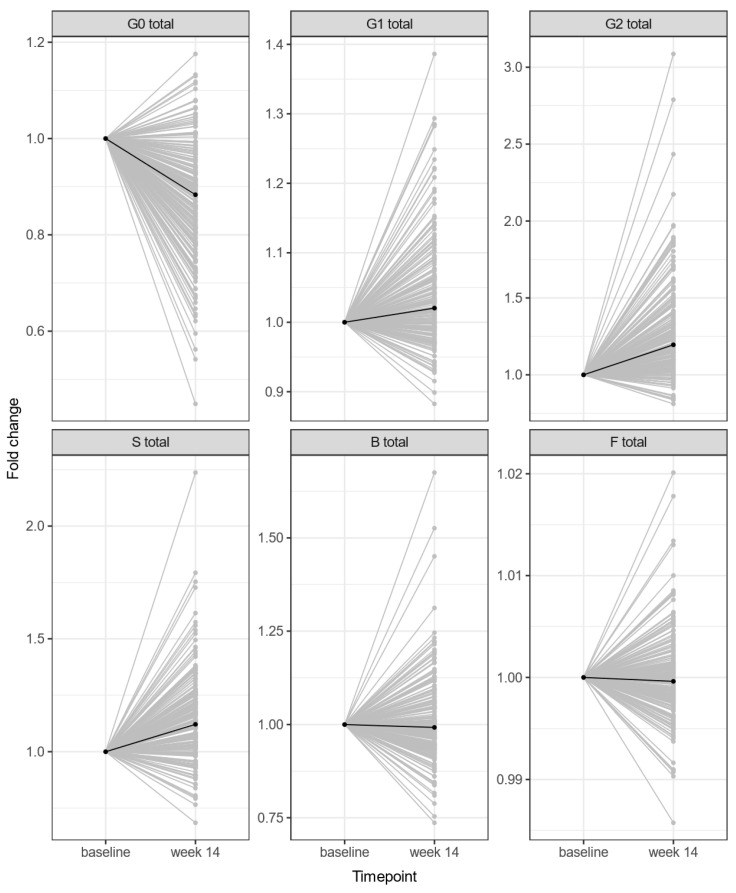
The change in derived IgG N-glycan traits during 14 weeks of anti-TNF treatment (both IFX/ADA). Median glycan values for each time point are bolded. *Y* axis: relative change in glycan value normalised to baseline; *X*-axis: time point (baseline, week 14). Abbreviations: G0—agalactosylated glycans, G1—monogalactosylated glycans, G2—digalactosylated glycans, S—mono- and disialylated glycans, B—glycans with bisecting GlcNAc, F—core-fucosylated glycans.

**Figure 2 biomolecules-13-00954-f002:**
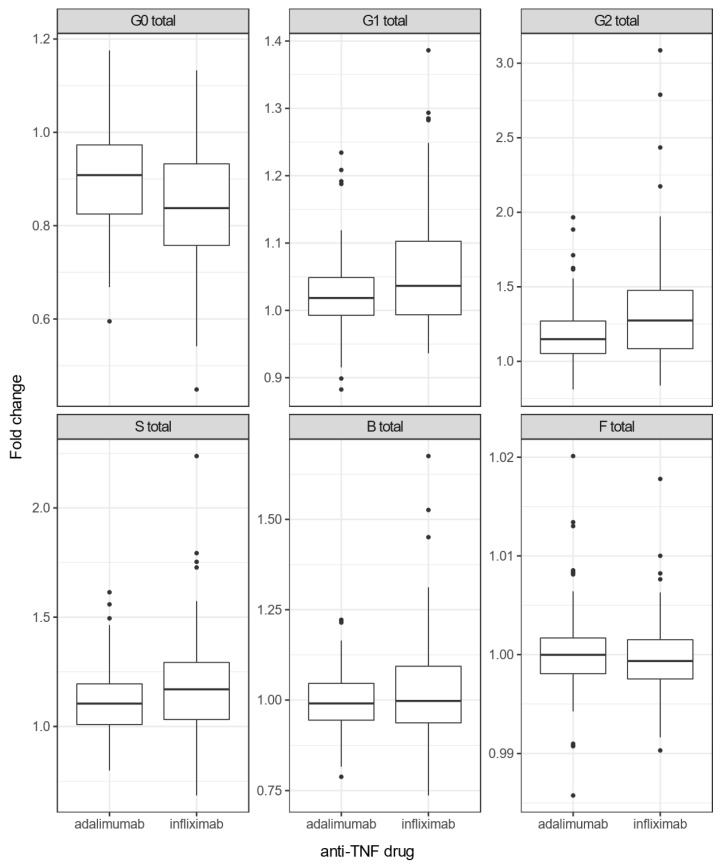
Change in derived IgG N-glycan traits in CD patients during the 14-week anti-TNF treatment with IFX or ADA. Each box represents the 25th to 75th percentiles (interquartile range-IQR). Lines inside boxes stand for the median. The whiskers are the lowest and highest values within boxes ± 1.5 × the IQR. Dots are outliers (>1.5 × IQR). Abbreviations: G0—agalactosylated glycans, G1—monogalactosylated glycans, G2—digalactosylated glycans, S—mono- and disialylated glycans, B—glycans with bisecting GlcNAc, F—core-fucosylated glycans.

**Figure 3 biomolecules-13-00954-f003:**
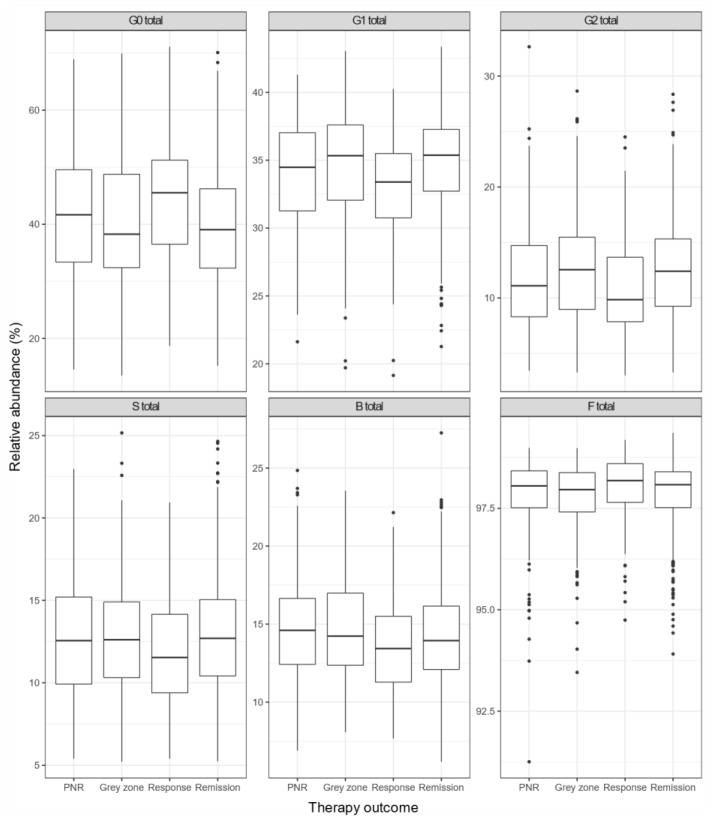
The difference in the relative abundance of derived IgG N-glycan traits at baseline between the groups of Crohn’s disease patients with different therapy outcomes (established at week 14). Each box represents the 25th to 75th percentiles (interquartile range—IQR). Lines inside boxes stand for the median. The whiskers are the lowest and highest values within boxes ± 1.5 × the IQR. Dots are outliers (>1.5 IQR). Abbreviations: PNR—primary non-responders to IFX/ADA therapy, Response—responders to IFX/ADA therapy, Grey zone—CD patients with an intermediate response between primary non-response and response, Remission—CD patients in remission. G0—agalactosylated glycans, G1—monogalactosylated glycans, G2—digalactosylated glycans, S—mono- and disialylated glycans, B—glycans with bisecting GlcNAc, F—core-fucosylated glycans.

**Table 1 biomolecules-13-00954-t001:** Demographics of the studied cohort.

	Week 0 (Baseline)	Week 14
Number of samples (N)	1315	198
Median age at first dose [IQR], yr	33 [23–47]	33 [24–46]
Number of females (%)	673 (51.2)	86 (43.4)
Median disease duration at first dose [IQR], yr	2.5 [0.7–9.0]	2.0 [0.6–10.1]
Anti-TNFα treatment (n)		
Infliximab (IFX)	820	100
Adalimumab (ADA)	495	98
Therapy outcome (n)		
Primary non-response (PNR)	258	33
Response	180	165
Grey zone	231	/
Remission	515	/
N/A	131	/

Week 0: first time point/baseline data collected before the first dose of anti-TNF therapy; Week 14—second time point data collected for the subset of patients immediately before the next scheduled anti-TNF injection/infusion.

**Table 2 biomolecules-13-00954-t002:** Change in derived IgG traits during the 14-week anti-TNF treatment (both IFX/ADA included) of CD patients. Analysis was performed by implementing a linear mixed-effects model, with time as a fixed effect and the individual sample measurement as a random effect. *p*-values were adjusted for multiple testing and considered significant if <0.05 (bold). Effect: model coefficient (slope) represents the change of a derived trait between two time points (expressed in standard deviation units).

Derived Trait	Effect	Standard Error	*p*-Value	Adjusted *p*-Value
G0	**−0.55**	0.04	1.34 × 10^−29^	**4.03 × 10^−29^**
G1	**0.35**	0.05	1.74 × 10^−12^	**2.61 × 10^−12^**
G2	**0.60**	0.04	1.51 × 10^−32^	**9.07 × 10^−32^**
S	**0.46**	0.04	1.88 × 10^−21^	**3.75 × 10^−21^**
B	0.02	0.04	6.25 × 10^−1^	6.25 × 10^−1^
F	−0.02	0.04	5.77 × 10^−1^	6.25 × 10^−1^

Abbreviations: G0—agalactosylated glycans, G1—monogalactosylated glycans, G2—digalactosylated glycans, S—mono- and disialylated glycans, B—glycans with bisecting GlcNAc, F—core-fucosylated glycans.

**Table 3 biomolecules-13-00954-t003:** Change in derived IgG N-glycan traits during the 14-week anti-TNF treatment between the groups of CD patients treated with ADA or IFX. Analysis was performed by implementing a linear mixed-effect model. Effect: the difference between two model coefficients (slopes), where each coefficient represents a treatment-specific change of a derived trait between two time points (expressed in standard deviation units). *p*-values were adjusted for multiple testing and considered significant if <0.05 (bold).

Derived Trait	Effect	Standard Error	*p*-Value	Adjusted *p*-Value
G0	**−0.24**	0.08	1.73 × 10^−3^	**3.46 × 10^−3^**
G1	**0.28**	0.09	1.08 × 10^−3^	**3.23 × 10^−3^**
G2	**0.26**	0.08	8.85 × 10^−4^	**3.23 × 10^−3^**
S	**0.18**	0.08	2.34 × 10^−2^	**3.51 × 10^−2^**
B	0.05	0.07	4.88 × 10^−1^	4.88 × 10^−1^
F	−0.06	0.08	4.73 × 10^−1^	4.88 × 10^−1^

Abbreviations: G0—agalactosylated glycans, G1—monogalactosylated glycans, G2—digalactosylated glycans, S—mono- and disialylated glycans, B—glycans with bisecting GlcNAc, F—core-fucosylated glycans.

**Table 4 biomolecules-13-00954-t004:** Associations between derived IgG N-glycan traits at baseline and therapy outcomes established at week 14 in PANTS cohort. *p*-values were adjusted for multiple testing and considered significant if <0.05 (bold).

Derived Trait	*p*-Value	Adjusted *p*-Value
G0	**9.81 × 10^−8^**	**1.96 × 10^−7^**
G1	**2.95 × 10^−8^**	**1.77 × 10^−7^**
G2	**7.98 × 10^−8^**	**1.96 × 10^−7^**
S	**1.31 × 10^−3^**	**1.97 × 10^−3^**
B	9.16 × 10^−2^	9.16 × 10^−2^
F	7.41 × 10^−2^	8.89 × 10^−2^

Abbreviations: G0—agalactosylated glycans, G1—monogalactosylated glycans, G2—digalactosylated glycans, S—mono- and disialylated glycans, B—glycans with bisecting GlcNAc, F—core-fucosylated glycans.

## Data Availability

The data presented in this study are available upon request from the corresponding authors.

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
