# Peer review of "Anti-TNF Biologicals Enhance the Anti-Inflammatory Properties of IgG N-Glycome in Crohn’s Disease"

_biomolecules, 2023, doi:10.3390/biom13060954_

Round 1

Reviewer 1 Report

Dear Authors,

Thank you for the opportunity to review this interesting paper. The topic is very interesting, especially for the scientific community with emphasis on Inflammatory Bowel Disease and biologicals. The Introduction is clearly written, Materials and Methods section is clear and it's easy to understand, Results sections is nicely presented and Discussion section supports the findings. In my own opinion, the paper can be accepted in the present form.

With kind regards,

Reviewer

Author Response

We thank Reviewer 1 for the positive feedback and kind comments.

Reviewer 2 Report

The submission of Hanić and co-workers compares IgG glycosylation of Crohn´s disease patients before and briefly after treatment with anti-TNF mAbs that should dampen inflammatory conditions.

A remarkable fraction of patients, however, does not respond. The aim of the current work was to detect changes of IgG glycosylation that would in any be associated with treatment outcome. To this end the authors applied their well-tried high-end methodology analyzing the N-glycans by UPLC with fluorescence detection and evaluating the data by – as far as I can rate that – appropriate statistics.

As so often, however, the observed effects are – although statistically significant, very small.
Moreover, they are by and large about what is expected to happen upon dampening of an inflammatory status.

Nevertheless, I would see the work published.
Nevertheless, the second, I would like to have answers on these questions:

à As the effect size is small, the question arises, if the authors took care of randomizing sample sequences when running the analysis?

à Table 1 shows extremely different numbers between Week 0 and Week 14. How can I understand these numbers? I assume that the 189 were those for which data for both time points became available? How and why did this sub-group emerge from the parent population.
This strange detail (is it a detail?) of the study should by no means be obscured.

à To grant a barrier-free access for non-clinicians the term “at baseline” could somewhen translated with “before start of treatment” and “primary non-responders” with “patients not responding to the first dose of anti-TNF antibody” – or so.

à for me as a semi or rather quarter statistician the meaning of Table 4 remains obscure. 4 conditions in Fig. 3 and one p-value? Line 397 tells me that the authors themselves could not interpret their result.

To conclude, a large effort to elucidate the glyco-aspects of CD treatment with a rather humble result that lies somewhere between the expectable and noise.

Author Response

Response to Reviewer 2 Comments

Point 1. As the effect size is small, the question arises, if the authors took care of randomizing sample sequences when running the analysis?

Response 1. We thank the Reviewer for the constructive comment. Indeed, we did perform the block randomisation of initial serum samples to minimize the effects of experimental variation and to maintain the same ratio of different groups (sex, age, therapy outcome, drug) across 18 plates in a 96-well format. Furthermore, we included a standard sample in six technical replicates per plate (replication) to assess quality control and allow batch correction on UHPLC IgG N-glycan data.

We add the following statement in line 123: “Prior to the IgG isolation, block randomisation was performed to define the position of samples across 96-well plates and replication standard samples were included as well. “

Point 2. Table 1 shows extremely different numbers between Week 0 and Week 14. How can I understand these numbers? I assume that the 189 were those for which data for both time points became available? How and why did this sub-group emerge from the parent population.
This strange detail (is it a detail?) of the study should by no means be obscured.

Response 2. We thank the Reviewer for pointing out the confusing table layout. The assumption is correct; samples at the second time point (week 14) were available for a smaller subset of CD patients, more precisely 198 of them.

Therefore, we have modified the statement in the Materials and Methods section in line 102 as follows: “A total of 1513 CD patients' serum samples were collected at two time points, before anti-TNF induction (week 0/baseline, N=1315), and for the subset of patients, immediately before the next scheduled anti-TNF injection/infusion (week 14, N=198).

Also, we have modified the Table 1. description including the following explanations in line 230: “Week 0: first time point/baseline data collected before the first dose of anti-TNF therapy; Week 14 – second time point data collected for the subset of patients immediately before the next scheduled anti-TNF injection/infusion.” Additionally, we have now provided Table 1 with the number of CD patients per group regarding therapy outcomes.

Point 3. To grant a barrier-free access for non-clinicians the term “at baseline” could somewhen translated with “before start of treatment” and “primary non-responders” with “patients not responding to the first dose of anti-TNF antibody” – or so.

Response 3. We thank the Reviewer for the comment. Our goal was to keep the article comprehensible to the scientists/readers outside this area as much as possible.

We have briefly explained the PNR abbreviation in the Introduction section at line 79, which is now modified as follows: “Up to one-third of patients do not respond to anti-TNF therapy induction regimen (primary non-response, PNR),…“.

In the Materials and Methods section in line 103, we explained that the term baseline or week 0 refers to the time point in which samples were collected before the anti-TNF induction and now is further clarified: „ … before the first dose of anti-TNF therapy (week 0/baseline, N=1315)…“

Point 4. for me as a semi or rather quarter statistician the meaning of Table 4 remains obscure. 4 conditions in Fig. 3 and one p-value? Line 397 tells me that the authors themselves could not interpret their result.

Response 4. We thank the Reviewer for the constructive comment. Table 4. shows the general association of a particular derived trait with the therapy outcomes, which is the reason for only one p-value per trait.

Now, we have provided an additional supplementary Table S1. to present a pairwise comparison of groups of CD patients with different therapy outcomes to show the differences of derived IgG traits at baseline.

However, we could not detect any glycan pattern that will enable us to detect PNR before the first dose of anti-TNF therapy. We could not detect any meaningful trend in the change of a particular derived trait either. For example, one could expect that the relative abundance of a particular trait for the Grey zone would be positioned somewhere between Non-response and Response, which was not the case in our dataset.